# Modeling the Reduction and Cross-Contamination of *Salmonella* in Poultry Chilling Process in China

**DOI:** 10.3390/microorganisms7100448

**Published:** 2019-10-13

**Authors:** Xingning Xiao, Wen Wang, Jianmin Zhang, Ming Liao, Hua Yang, Weihuan Fang, Yanbin Li

**Affiliations:** 1College of Biosystems Engineering and Food Science, Zhejiang University, Hangzhou 310058, China; xingningxiao@126.com; 2State Key Laboratory for Quality and Safety of Agro-products, MOA Laboratory of Quality & Safety Risk Assessment for Agro-products (Hangzhou), Institute of Quality and Standard of Agricultural Products, Zhejiang Academy of Agricultural Sciences, Hangzhou 310021, China; yanghua@mail.zaas.ac.cn; 3College of Veterinary Medicine, South China Agricultural University, Guangzhou 510642, China; junfeng-v@163.com (J.Z.); mliao@scau.edu.cn (M.L.); 4College of Animal Sciences, Zhejiang University, Hangzhou 310058, China; whfang@zju.edu.cn; 5Department of Biological & Agricultural Engineering, University of Arkansas, Fayetteville, Arkansas, AR 72701, USA

**Keywords:** modeling, *Salmonella*, reduction, cross-contamination, poultry chilling

## Abstract

The study was to establish a predictive model for reduction and cross-contamination of *Salmonella* on chicken in chilling process. Reduction of *Salmonella* on chicken was 0.75 ± 0.04, 0.74 ± 0.08, and 0.79 ± 0.07 log CFU/g with 20, 50, and 100 mg/L of chlorine, respectively. No significant differences of bacterial reductions with 20–100 mg/L of chlorine were found and a Normal (−0.75, 0.1) distribution could describe the uncertainty of bacterial reductions. Inoculated and non-inoculated chicken samples were washed together and bacterial transfer rates among them were 0.13%–0.004% with 20–100 mg/L of chlorine. No significant differences of transfer rates with 50–100 mg/L of chlorine were observed and a Triangle (−2.5, −1.5, −1.1) distribution could describe the log transfer rate. Additionally, a 3-factor response surface model based on the central composite design was developed to evaluate the effects of initial contamination level (1–5 log CFU/g), pre-chill incidence (3%–40%) and chlorine concentration (0–100 mg/L) on post-chill incidence. The post-chill incidences in these treatments were within 30%–91.7%. The developed model showed a satisfactory performance to predict the post-chill incidence as evidenced by statistical indices (pseudo-*R*^2^ = 0.9; *p* < 0.0001; RMSE = 0.21) and external validation parameters (*B_f_* = 1.02; *A_f_* = 1.11).

## 1. Introduction

Microbial contamination is a widespread public concern in the meat industry because it can shorten the shelf life and increase the safety risk of fresh meat and meat products [1,2]. Recently, contamination of *Salmonella* continues to be a major concern in poultry industry [3,4]. It was found that non-typhoid *Salmonella* annually caused 9.87 million gastroenteritis cases in China and more than half of the retail chicken carcasses were contaminated with *Salmonella* [4,5]. *Salmonella* from the skin, feather, spillage of intestinal, and water are disseminated from carcass to carcass as they are moved down the poultry slaughtering line [6]. Incoming animal-associated contamination of the abattoir environment as well as the slaughter of large numbers of animals on the same slaughter line contributes to direct contamination or cross-contamination of chicken carcasses during slaughtering.

Following poultry evisceration, the next step in poultry processing is chilling by adding chlorine into the water to reduce microbial contamination and cross-contamination [6,7,8]. Chlorine-based decontamination disinfectants, such as sodium hypochlorite, chlorine dioxide, acidified sodium chlorite, and monochloramine, are the most commonly used sanitizers in poultry slaughterhouses [9,10,11]. Chlorine concentrations used in poultry slaughterhouses are different from country to country. The U.S. Department of Agriculture Food Safety and Inspection Service (USDA-FSIS) requires that processors add 20 to 50 mg/L of chlorine to chilling water to reduce pathogens and prevent pathogen cross-contamination of poultry carcasses [12]. In China, 50 to 100 mg/L of chlorine is commonly used in poultry chilling process [13]. Although other antimicrobial agents are approved for poultry processing such as organic acids, chlorine still remains the most widely used antimicrobial chemical by the poultry industry due to its antimicrobial efficacy, convenience, and low price [8,14,15].

The chilling tank is a high-risk area where cross-contamination between contaminated and non-contaminated carcasses occurred via washing water, leading the changes of bacterial contamination level and incidence [6]. A higher incidence of *Salmonella* was observed in poultry carcasses from post-chill than those from pre-chill [16,17]. Several factors have been determined to be involved with bacterial incidence during chilling, including poultry contamination level, chlorine concentration and seasonal effect [6,16,18]. Predictive microbiology is an important tool for investigating the behavior of pathogens under prescribed environmental conditions through the development of mathematical models [19,20]. In previously studies, a Weibull model was used to describe bacterial reduction and a logistic model was developed to predict post-chill incidence in poultry chilling process with 0–50 mg/L of chlorine, while chlorine concentration with 50–100 mg/L is common used in China according to the national performance standard of the operation procedure for poultry slaughtering [6,21,22]. Very few predictive models are available for the description of *Salmonella* reduction and cross-contamination during chilling for quantitative microbial risk assessment (QMRA) of the poultry supply chain in China.

Therefore, the objectives of this study were: (i) To investigate the effect of chlorine concentration on the reduction and transfer of *Salmonella* in chilling process; (ii) to determine the post-chill incidence of *Salmonella* on chicken products with different initial contamination levels, pre-chill incidences and chlorine concentrations; and (iii) to develop probability distribution and predictive models for describing the bacterial reduction, bacterial transfer rate and bacterial post-chill incidence. The developed predictive models in this study were critical to provide reliable inputs to a QMRA model for the whole poultry supply chain in China.

## 2. Materials and Methods

### 2.1. Bacterial Inoculum

Five serovars of *Salmonella* (Stanley BYC12, Indiana HZC10, Typhimurium YXC1, Thompson LWC10, Kentucky CBC2) isolated from poultry slaughterhouses (Guangzhou, Guangdong, China) in China were used in this study. The bacterial strains were maintained in brain heart infusion broth (BHI, Becton Dickinson (BD), Franklin Lakes, NJ, USA) containing 20% (*v*/*v*) glycerol at −80 °C. Each strain was separately incubated in BHI at 37 °C for 24 h and cultured to approximately 9 log CFU/mL. Equal volumes of each culture suspension were mixed to obtain a five-strain mixture of *Salmonella*. Appropriate 10-fold dilutions in sterile phosphate buffered saline (PBS, Sigma, St. Louis, MO, USA) were made and plated on xylose lysine Tergitol 4 (XLT4, BD) agar to determine the cell number in the inoculums.

### 2.2. Preparation and Inoculation of Chicken Samples

Skin injuries will happen due to the weak skin is more susceptible to mechanical tears, leading to skinless chicken meat exposure to the chilling water, especially in small scale slaughterhouses in China. In the preliminary test, bacterial reduction and bacterial transfer rate on skin-on chicken wingettes and skinless chicken breasts were compared, which was not significantly different (*p* > 0.05) as determined by analysis of variance (ANOVA). Therefore, chicken breast samples (weight: 25 ± 1 g, size: 5 × 3 × 2 cm) were used in the tests, which were purchased from a local supermarket (Hangzhou, Zhejiang, China) and stored in a freezer at −20 °C. After thawing overnight at 4 °C, the samples were exposed to UV light for 30 min in a biosafety cabinet (Thermo Fisher 1389, Waltham, MA, USA) to decontaminate initial microbial contamination. Controls in the test showed that no *Salmonella* was initially present on the chicken breast samples. Then the chicken breast samples were submerged into *Salmonella* suspensions for 30 min and transferred to plastic plates for another 30 min to allow bacterial full attachment, both of which were performed at ambient temperature (25 ± 1 °C).

### 2.3. Chilling Treatments

The chilling conditions were designed based on the operation of poultry slaughterhouses (Guangzhou, Guangdong) in China, the ratio of chicken samples to chilling water was 1:10 (*w/v*). Chilling treatments at 4 °C were carried out in a laboratory water bath (TX150, Grant, Royston, UK) equipped with a digital thermometer (34970A, Agilent, Santa Clara, CA, USA) to monitor temperatures of water. Sodium hypochlorite (NaClO) stock solution containing 56.8 mg/mL chlorine was purchased from Sangon Biotech Co., Ltd., Shanghai, China. Chlorination of chilling water was prepared by diluting NaClO stock solution using sterile Milli-Q water (Millipore, Billerica, MA, USA) and the chlorine concentration in chilling water was determined using a Palintest ChlorSense meter (CS 100, Gateshead, Tyne and Wear, UK).

To evaluate the effect of chlorine concentrations (0, 20, 50, and 100 mg/L) on the bacterial reduction, twelve inoculated chicken breast samples were submerged into the water bath. Three inoculated samples were removed for processing every 10 min within a 40 min treatment period. The samples were individually added to sterile stomacher bags (Seward, London, UK) containing 25 mL buffered peptone water (BPW, BD) and homogenized for 1 min in a Model 400 food stomacher (Seward, London, UK). The homogenates were diluted to appropriate concentrations for bacterial enumeration. The initial inoculation load of chicken breasts was 5.6 ± 0.1 log CFU/g. The bacterial population on chicken samples at *t* = 0 was determined using the inoculated samples without treatment. Each treatment was repeated three times on different days and duplicated plates were used in microbial tests for each sample.

To assess the effect of chlorine concentrations (0, 20, 50, and 100 mg/L) on the bacterial transfer rate from inoculated chicken breasts to non-inoculated chicken breasts via chilling water, twelve inoculated and twelve non-inoculated chicken breasts were submerged into the water bath together. Three inoculated and three non-inoculated samples were removed for processing every 10 min within a 40 min treatment period. The non-inoculated samples were individually added to sterile stomacher bags containing 25 mL of BPW and homogenized for 1 min in a Model 400 food stomacher. The homogenates were diluted to appropriate concentrations for bacterial enumeration. The initial inoculation load of chicken breast samples was 5.6 ± 0.1 log CFU/g. The bacterial population on chicken samples at *t* = 0 was determined using the inoculated samples without treatment. Each treatment was repeated three times on different days and duplicated plates were used in microbial tests for each sample.

To evaluate the combined effects of initial contamination levels (1, 2, 3, 4, and 5 log CFU/g), pre-chill incidences (3%, 10.2%, 21.5%, 32.8%, and 40%) and chlorine concentrations (0, 20, 50, 80, and 100 mg/L) on post-chill incidence of *Salmonella*, a response surface model based on the central composite design was developed with the JMP10 software (SAS Institute, Cary, NC, USA) to predict the post-chill incidence. In multivariable analyses, the traditional optimization technique of changing one variable at a time to study the variable response effect is impracticable and does not represent the interaction effect between different factors. Therefore, experimental statistical design considers one of the most useful methods in obtaining valuable and statistically significant models of a phenomenon by performing a minimum number of calculated experiments. It also considers interactions among the variables and can be used to optimize the operating parameters in multivariable analyses. Response surface methodology was used for the modeling and analysis of problems in which a response of interest is influenced by several variables to optimize the same response. A favorite model in response surface methodology is the central composite design, which is efficient and flexible in providing adequate data on the effects of variables and overall experiment error even with a fewer number of experiments [23,24,25]. Tested variables (initial contamination level, pre-chill incidence, and chlorine concentration) were denoted as *X*_1_, *X*_2_, and *X*_3_, respectively, and each factor in the design was studied at five different levels (−α, −1, 0, +1, +α) (Table 1), with a total of 20 runs according to the experimental design (Table 2). Among the 30 chicken breasts, a total of one, three, seven, ten, and twelve chicken breasts were inoculated to obtain a pre-chill incidence of 3%, 10.2%, 21.5%, 32.8%, and 40%, respectively. Then, inoculated and non-inoculated samples were mixed together in the water bath for a 40 min chilling treatment. The samples were individually added to sterile stomacher bags containing 25 mL of BPW and homogenized for 1 min in a Model 400 food stomacher. The homogenates were diluted to appropriate concentrations for bacterial enumeration. The whole experiment was repeated twice on different days and duplicated plates were used in microbial tests for each chicken breast sample.

### 2.4. Bacterial Enumeration

In the preliminary test, the recovery of injured *Salmonella* was considered. XLT4 agar was compared with Tryptic Soy agar (TSA, BD) in plate counting and only less than 0.1 log CFU/g difference was observed, which was not significantly different (*p* > 0.05) as determined by ANOVA. Therefore, populations of *Salmonella* were selectively enumerated on XLT4 agar. The homogenates were serially 10-fold diluted in BPW and a 50 µL portion of appropriate dilutions was plated in duplicate onto the XLT4 agar using a spiral plater (WASP 2, Don Whitley Scientific, Shipley, UK). The plates were incubated at 37 °C for 18 h. Colonies on XLT4 agar plates were enumerated by a ProtoCOL 3 automated colony counter (Synbiosis, Cambridge, UK). The limit of detection was one colony for 50 μL sample (1.3 log CFU/mL). The viable bacterial populations on samples were expressed as CFU/g.

When the bacterial population on samples were under the detection limit, enrichment test was conducted to determine whether there were *Salmonella* survivors in samples. One milliliter of homogenates and water sample were aseptically collected in 9 mL of Selenite Cystine Broth (SC, HopeBiol, Qing Dao, Shan Dong, China) and incubated at 37 °C for 24 h, respectively. The enriched samples were plated onto the XLT4 agar and incubated at 37 °C for 18 h. Black colonies on the XLT4 agar denoted that *Salmonella* survivors in samples.

### 2.5. Calculation of Bacterial Reduction, Bacterial Transfer Rate and Post-chill Incidence

Counts of surviving bacteria were log transformed, and the bacterial reduction (*Y_red_*) in bacterial population was calculated with the Equation (1):(1)$$Y_{red}=\log N_{t}-\log N_{0}$$
where *N_t_* (CFU/g) is the number of bacteria at time *t* (min) and *N*_0_ (CFU/g) is the initial number of bacteria.

The transfer rate (TR%) was defined as the percentage of bacterial transferred from the donor surface to recipient surface, which could be calculated by Equation (2).
(2)$$TR\left(\%\right)=\frac{N_{recipient}}{N_{donor}}\times 100$$
where, *N_recipient_* and *N_donor_* are bacterial population on the recipient surface and donor surface, respectively. In this study, *N_recipient_* (CFU) is the number of *Salmonella* on non-inoculated chicken breast samples after chilling, and *N_donor_* (CFU) is the initial microbial load on inoculated chicken breast samples.

Positive contamination of *Salmonella* on each chicken breast was recorded. Post-chill incidence *(Y_pc_*) could be calculated by Equation (3):(3)$$Y_{pc}\left(\%\right)=\frac{A}{B}\times 100$$
where, *A* and *B* is number of positive chicken breasts and total number of chicken breasts, respectively.

### 2.6. Model Development

In the cases that no significant differences of bacterial reduction and transfer rate were observed, probability distribution model could provide all potential results [26]. Besides, bacterial reductions and transfer rates showed a large variation because of multiple uncertainties that were involved, as well as the inherent errors in microbial collection from surfaces and enumeration techniques. Hence, to describe the uncertainty, probability distributions were defined based on the Kolmogorov–Smirnov test to describe the uncertainty using @Risk 7.5 software (Palisade, Newfield, NY, USA).

The stepwise regression model was constructed recursively by adding or deleting one independent prediction at each time. A backward elimination was used which starts from a model with a full set of base functions and then gradually dropped out the predictor with the least effect in the model in a stepwise fashion [27]. For the prediction of post-chill incidence, a second order polynomial equation was fitted to the post-chill incidence by a backward stepwise regression. This resulted in an empirical model that related the response measured to the independent variables of the experiment. For a three-factor system the model is Equation (4):(4)$$Y_{pc}=K_{0}+K_{1}X_{1}+K_{2}X_{2}+K_{3}X_{3}+K_{4}X_{1}^{2}+K_{5}X_{2}^{2}+K_{6}X_{3}^{2}+K_{7}X_{1}X_{2}+K_{8}X_{1}X_{3}+K_{9}X_{2}X_{3}$$
where *Y_pc_* is the post-chill incidence (%); *X*_1_, *X*_2_, and *X*_3_ are initial contamination level (log CFU/g), pre-chill incidence (%) and chlorine concentration (mg/L), respectively. *K*_0_ is the intercept; *K*_1_, *K*_2_, and *K*_3_, are linear coefficients; *K*_4_, *K*_5_, and *K*_6_ are squared coefficients; *K*_7_, *K*_8_, and *K*_9_ are interaction coefficients.

### 2.7. Model Evaluation and Validation

ANOVA was used to evaluate significance and adequacy of the model. Fitting goodness of the model was characterized by correlation coefficient (*R*^2^) and the root mean square error (RMSE). Eight random independent trials were conducted with chicken carcasses to calculate the bias factors (*B_f_*) (Equation (5)) and accuracy factors (*A_f_*) (Equation (6)) for model validation (Table 3) [28]. The selected parameters were within the original range of the experimental design but not included in the development of the model [29].
(5)$$B_{f}=10^{\left[\sum_{i=1}^{n}\log\left(\frac{obs}{pred}/n\right)\right]}$$
(6)$$A_{f}=10^{\left[\sum_{i=1}^{n}\left|\log\left(\frac{obs}{pred}\right)\right|/n\right]}$$
where *n* is the number of trials, *obs* is the observed values of post-chill incidence (%), and *pred* is the predicted values of post-chill incidence (%).

### 2.8. Color Measurements

Color changes of chicken breasts that occurred during chilling with 0, 20, 50, and 100 mg/L of chlorine were determined using a Chroma Meter CR 400 instrument (Minolta, Osaka, Japan). Samples were measured after a 40 min treatment. Values of *L, a*, and *b* represented the lightness, redness and yellowness, respectively, were recorded. The total color difference (Δ*E*) was calculated according to Equation (7). Δ*E* values of 1 to 2 mean that color change is perceived through close observation, and values of 3 to 10 mean changes is perceived at a glance [30]. All measurements were taken on five sites of each chicken breast sample.
(7)$$∆E=\sqrt{∆L^{2}+∆a^{2}+∆b^{2}}$$
where Δ*L*, Δ*a*, and Δ*b* are the differences of lightness, redness, and yellowness between the treated and untreated samples, respectively.

### 2.9. Transmission Electron Microscopy (TEM)

In order to observe the morphology differences of the *Salmonella* cells with 0, 50, and 100 mg/L of chlorine, ultrastructural changes of bacterial cells were observed with a transmission electron microscope (TEM, Hitachi 7650, Ibaraki, Japan). The treated samples were removed from the water bath and placed into a sterile stomacher bag containing 25 mL BPW and then squeezed by hand to wash off attached bacteria. The wash solution for each treatment was collected into a 50 mL centrifuge tube and centrifuged at 6000× *g* for 10 min. The bacterial pellets were transferred into sterile 1.5 mL centrifuge tubes. Samples were fixed with 1 mL of 2.5% (*v/v*) glutaraldehyde (Sangon Biotech, Shanghai, China) and examined using a TEM as previously described [31].

## 3. Results and Discussion

### 3.1. Salmonella Reduction at Different Chlorine Concentrations

In non-chlorinated water, bacterial reductions of *Salmonella* on chicken breasts after the 40 min treatments were 0.49 ± 0.1 log CFU/g. The reductions of *Salmonella* on chicken breasts after the 40 min treatments were 0.75 ± 0.04, 0.74 ± 0.08, and 0.79 ± 0.07 log CFU/g with 20, 50, and 100 mg/L of chlorine, respectively (Figure 1). There were no significant differences of bacterial reductions among 20, 50, and 100 mg/L of chlorine and the treatment time was not a significant variable to inactivate the bacteria (*p* > 0.05).

The effectiveness of chlorine on reduction of *Salmonella* on chicken was limited [8,32]. It is believed that the chlorine does not easily access the bacteria in ridges and crevices on poultry carcasses [15,33]. Besides, *Salmonella* on chicken surface cannot be removed effectively by chlorine due to the interference of oil between the sanitizers and the surface [32]. Generally, treatments with chlorine resulted in less than a 1 log reduction on carcasses [15,34,35]. For example, Yang et al. (2010) [21] found that the reductions of *Salmonella* with 10–50 mg/L of chlorine were < 0.5 log CFU/cm^2^ in chicken skin. Lee et al. (2013) [15] observed that the combination of 200 mg/L of chlorine and ultrasound significantly reduced *Salmonella* populations by 0.5–0.8 log CFU/g in chicken skin. Tamblyn and Conner (1997) [36] found that chlorine levels must be at least 400 mg/L to kill the attached *Salmonella* on broilers. In this study, bacterial reductions on chicken breasts were less than 1 log CFU/g with 20–100 mg/L of chlorine, showing the effectiveness of chlorine on reduction of *Salmonella* was limited, which was consistent with previous studies.

### 3.2. Transfer of Salmonella at Different Chlorine Concentrations

The averages of TR% on chicken breasts at the end of 40 min treatments were 0.12, 0.08, 0.03, and 0.02% with 0, 20, 50, and 100 mg/L of chlorine, respectively. The average bacterial populations in chilling water at the end of 40 min treatments of 0, 20, 50, and 100 mg/L of chlorine were 4, 3.5, <1.3 (detection limit) and 0 log CFU/mL, respectively. Chlorine with a concentration of 0–20 mg/L failed to mitigate bacterial transfer due to a high contamination level in the washing water. Bacterial populations in water significantly decreased after chilling with 50 mg/L of chlorine and no *Salmonella* survivor in water was found at 100 mg/L of chlorine. The results indicated that 50–100 mg/L of chlorine were effective in controlling *Salmonella* transfer in chilling process. Chilling water is an ideal medium for the potential spread of bacterial pathogens during processing. Therefore, the presence of a sanitizing agent such as sodium hypochlorite in the wash water is critical to preventing pathogen survival and transfer [37,38]. Similarly, Mead et al. (1994) [39] reported that less than 30 mg/L of chlorine did not prevent microbial cross-contamination on poultry. Chlorination of chilling water was effective to control cross-contamination, but was limited for reducing the bacteria attached on the chicken carcasses [21,39].

### 3.3. Salmonella Post-Chill Incidence under Different Initial Contamination Levels, Pre-Chill Incidences, and Chlorine Concentrations

The post-chill incidences of *Salmonella* on chicken breasts under different initial contamination levels, pre-chill incidences, and chlorine concentrations are presented in Table 2. The effect of initial contamination level, pre-chill incidence and chlorine concentration on the post-chill incidence was significant (*p* < 0.05). In Table 2, with a 5 log CFU/g of *Salmonella* inoculated on chicken breasts, the post-chill incidence of *Salmonella* was 90%, but was reduced by 60% at initial contamination level of 1 log CFU/g. Besides, with the same initial contamination level and chlorine concentration, the post-chill incidence of *Salmonella* was 78.4% when washed at a 40% pre-chill incidence, but was reduced by 38.4% with a 3% pre-chill incidence. With the same initial contamination level and pre-chill incidence, the post-chill incidence of *Salmonella* was 81.7% in non-chlorinated chilling water, but was reduced to 23.3% after chilling with 50 mg/L of chlorine. Yang et al. (2002) [6] reported that with a 43.3% pre-chill contamination, the post-chill contamination of *Salmonella* was more than 90% without chlorination, but was reduced to 20% after chlorination of chill water with 50 mg/L of chlorine. Chlorination of chilling water could contribute significantly toward a reduction in *Salmonella* incidence on commercially processed carcasses [6,40]. However, when the pathogen has a resistance to chlorine, a result has been reported that no correlation between the incidence of *Salmonella* on post-chill poultry carcasses and the chlorine concentration in chilling process [16].

### 3.4. Model Development

There were not significant correlation between bacterial reductions with the 20–100 mg/L chlorine treatments and the chilling time (*p* > 0.05) A Normal (−0.75, 0.1) distribution could describe the bacterial reductions with 20–100 mg/L of chlorine (Figure 2). Normal distributions, fitted by the sampling data randomly collected before and after chilling in slaughterhouse, have been used to describe the *Campylobacter* reductions on chicken carcasses [41,42]. Besides, there were not significant correlation between bacterial transfer rates with the treatments of 50–100 mg/L of chlorine and the chilling time (*p* > 0.05) Log transfer rates were essentially normally distributed and a Triangle (−2.5, −1.5, −1.1) distribution could describe the TR with 50–100 mg/L of chlorine (Figure 3). Normal distributions have been used to describe the variation of TR in previous studies [43,44]. A paradigmatic study was reported by Chen et al. (2001) [43]. They investigated bacterial transfer rates between hands and other common surfaces involved in food preparation in kitchen, and found the distribution of the logarithmic transfer rates appears approximately normal.

A backward stepwise regression was carried out using JMP software to develop a simplified response surface model only with the significant variables. By using ANOVA on the estimated parameters of all variables, we found statistical significance with three linear coefficients (*X*_1_, *X*_2_ and *X*_3_) (*p* < 0.001), and the quadratic coefficient ($X_{3}^{2}$) (*p* < 0.05) for estimation of bacterial post-chill incidence. A summary of the estimated parameters for uncoded variables with significances is given in Table 4. The simplified model is shown as Equation (8).
(8)$$Y_{pc}=18.28+15.75X_{1}+0.757X_{2}-0.636X_{3}+0.0044X_{3}^{2}$$
where *Y_pc_* is post-chill incidence (%), *X*_1_ is the initial contamination level (log CFU/g), *X*_2_ is the pre-chill incidence (%), and *X*_3_ is the chlorine concentration (mg/L).

The response surface plot in Figure 4 demonstrates that the effect of initial contamination level, pre-chill incidence and chlorine concentration on the post-chill incidence of *Salmonella* in chilling process. The response surface plot in Figure 4A describes the post-chill incidence of initial contamination level and pre-chill incidence at the fixed chlorine concentration of 50 mg/L (coded level of 0). The post-chill incidence of *Salmonella* was significantly reduced (*p* < 0.05) as the initial contamination level and pre-chill incidence decreased, and the post-chill incidence was more sensitive to the initial contamination level (Figure 4A). Post-chill incidence was slightly sensitive to the change of chlorine concentration, as compared with initial contamination level and pre-chill incidence (Figure 4B,C). The goodness-of-fit of a predictive model has a satisfactory performance as evidenced by statistical indices (pseudo-*R*^2^ = 0.9; *p* < 0.0001; RMSE = 0.21).

### 3.5. Model Evaluation and Validation

The model has a good statistical performance as shown by pseudo-*R*^2^ (0.9), probability value (*p* < 0.0001) and the lack of fit test (*p* > 0.05). A statistical validation is insufficient to evaluate accuracy of the model to predict post-chill incidence. Therefore, an external validation was carried out. The results of eight independent experiments (not included in the model development) shown in Table 3 were used to calculate the *B_f_* and *A_f_* based on Equations (5) and (6). The calculated *B_f_* was 1.02, which lies in the acceptable range of 0.9–1.05 [28,45]. The *A_f_* value was 1.11, revealing a merely 11% difference between the observations and predictions. These results indicated that the developed model could give a reliable prediction for post-chill incidence within the range of variables employed.

### 3.6. Color Changes during Chilling

For poultry products, color and visual appearance are important attributes to consumers [46]. The changes of color (Δ*L*, Δ*a*, Δ*b*) and Δ*E* values of the treated chicken breasts are summarized in Table 5. The *L*, *a* and *b* value of the chicken breasts color did not change with the treatments (*p* > 0.05). There were no obvious color changes of chicken breasts after chilling with 20–100 mg/L of chlorine (Δ*E* < 10). In general, the color changes of chicken breasts were acceptable based on the visual inspection.

### 3.7. Morphological Changes Reveled by TEM

We closely examined the morphology changes of bacterial cells using TEM. *Salmonella* cells in the absence of chlorine treatment retained their round shapes (Figure 5A). However, after chlorine treatment, the cells were completely disrupted and leaked cytoplasmic material to the extracellular medium (Figure 5B,C). Chlorine reacts with water to form the active antimicrobial hypochlorous acid, which destroys microbial cells by hindering carbohydrate metabolism [46].

## 4. Conclusions

Chlorine with concentrations of 50–100 mg/L was effective for reducing bacteria in chilling water to control cross-contamination, but was limited for reducing the bacteria attached on the chicken breasts. A Normal (−0.75, 0.1) distribution model could describe the bacterial reduction in chilling process and a Triangle (−2.5, −1.5, −1.1) distribution could describe the logarithm of transfer rate. For prediction of post-chill incidence, the developed response surface model has a satisfactory performance, which could be used to provide the input for QMRA model of *Salmonella* in poultry supply chain in China.

## Figures and Tables

**Figure 1 microorganisms-07-00448-f001:**
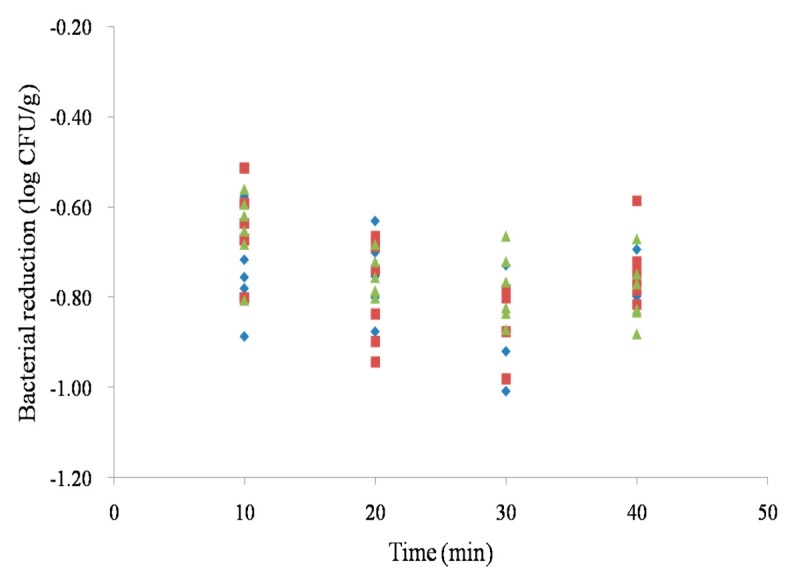
Bacterial reductions of *Salmonella* on chicken breasts with 20 (◆), 50 (■) and 100 (▲) mg/L of chlorine.

**Figure 2 microorganisms-07-00448-f002:**
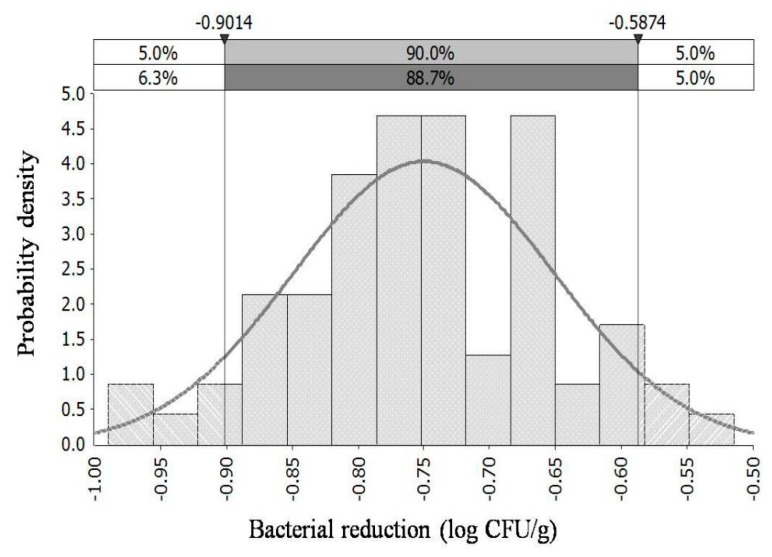
Probability distribution of *Salmonella* reduction on chicken breasts in chilling water with 20–100 mg/L of chlorine.

**Figure 3 microorganisms-07-00448-f003:**
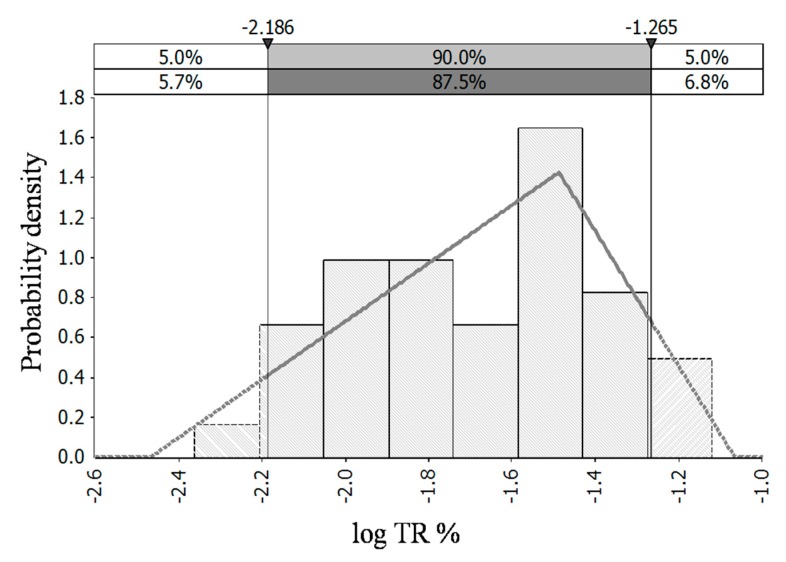
Probability distribution of transfer rate (log%) among chicken breasts in chilling water with 50–100 mg/L of chlorine.

**Figure 4 microorganisms-07-00448-f004:**
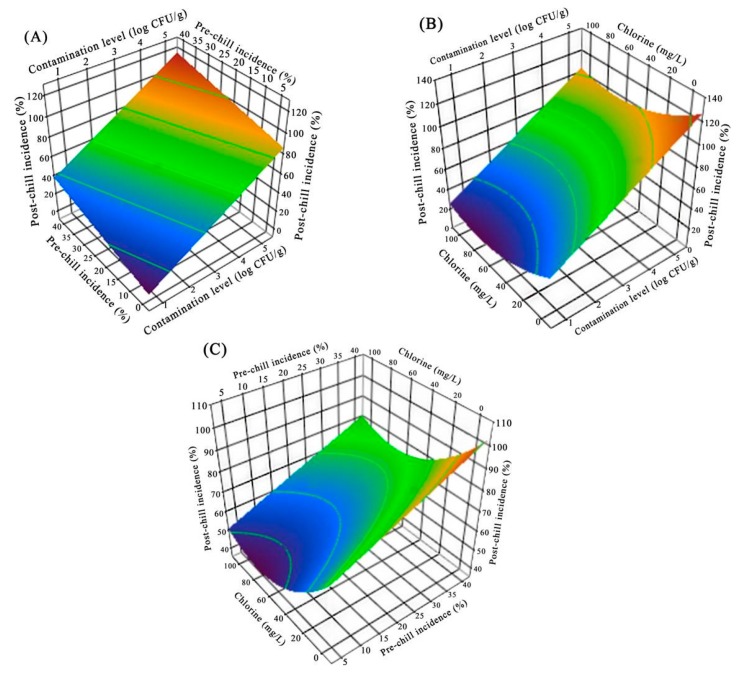
Response surface plots describing the effect of initial contamination level and pre-chill incidence (**A**); initial contamination level and chlorine concentration (**B**); and pre-chill incidence and chlorine concentration (**C**) on post-chill incidence of *Salmonella* in chilling process.

**Figure 5 microorganisms-07-00448-f005:**
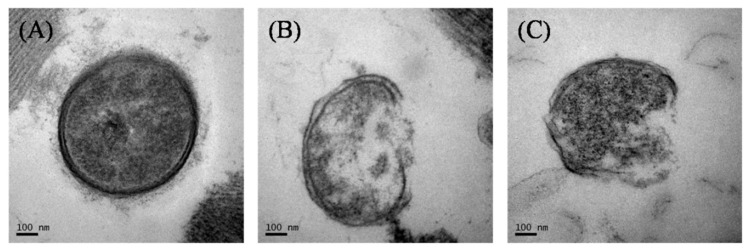
TEM images of *Salmonella* treated with 0 (**A**), 50 (**B**), and 100 mg/L (**C**) of chlorine.

**Table 1 microorganisms-07-00448-t001:** Levels of variables in the experiment.

Variable	Range	Level				
		−α	−1	0	1	*α*
Initial contamination level (log CFU/g)	1–5	1	2	3	4	5
Pre-chill incidence (%)	3–40	3	10.2	21.5	32.8	40
Chlorine (mg/L)	0–100	0	20	50	80	100

**Table 2 microorganisms-07-00448-t002:** Experimental design and post-chill incidence of *Salmonella* in chilling process.

Run	Initial Contamination Level (log CFU/g)	Pre-Chill Incidence (%)	Chlorine (mg/L)	^a^ Post-Chill Incidence (%)
1	2	32.8	80	43.3 ± 14.1*^abcd^*
2	4	10.2	80	66. 7 ± 4.7*^abc^*
3	3	21.5	50	65 ± 2.4*^abc^*
4	2	10.2	20	40 ± 12.4*^bcd^*
5	3	21.5	50	68.4 ± 11.8*^ab^*
6	4	32.8	20	91.7 ± 7.1*^a^*
7	3	21.5	50	65 ± 11.8*^abc^*
8	4	32.8	80	85 ± 2.4*^a^*
9	2	10.2	80	35 ± 7.1*^bcd^*
10	2	32.8	20	60 ± 28.3*^bcd^*
11	3	21.5	50	70 ± 14.1*^ab^*
12	4	10.2	20	90 ± 9.4*^a^*
13	3	21.5	0	81.7 ± 21.2*^a^*
14	3	40	50	78.4 ± 25.9*^a^*
15	5	21.5	50	90 ± 4.7*^a^*
16	3	21.5	50	65 ± 4.7*^abc^*
17	3	3	50	40 ± 18.9*^bcd^*
18	3	21.5	100	65 ± 2.4*^bc^*
19	3	21.5	50	63.3 ± 4.7*^abc^*
20	1	21.5	50	30 ± 4.7*^cd^*

^a^ Values are means ± standard deviations (*n* = 2). Values followed by different superscript letters are significantly different (*p* < 0.05).

**Table 3 microorganisms-07-00448-t003:** Eight independent trials to validate the model for post-chill incidence in chilling process.

Run	Initial Contamination Level (log CFU/g)	Pre-Chill Incidence (%)	Chlorine (mg/L)	Post-Chill Incidence (%)	
				Observed	Predicted
1	3	12.5	10	62.5 ± 8.8	69.1
2	3	37.5	10	100.0 ± 0.0	88.0
3	3	12.5	70	50.0 ± 0.0	52.0
4	3	37.5	70	87.5 ± 8.8	71.0
5	5	12.5	10	100.0 ± 0.0	100.6
6	5	37.5	10	100.0 ± 0.0	119.5
7	5	12.5	70	100.0 ± 0.0	83.5
8	5	37.5	70	100.0 ± 0.0	102.5

**Table 4 microorganisms-07-00448-t004:** The estimated parameters for variables with significance.

Factor	Value	Standard Error	Prob > *F*
Intercept	92.0	3.3	
*X* _1_	31.0	2.8	< 0.0001 **
*X* _2_	13.6	2.8	0.0002 *
*X* _3_	−9.7	2.8	0.004 *
*X* _3_ ^2^	10.8	4.6	0.03 *

Asterisks indicate statistical significance by *F* test (** *p* < 0.0001; * *p* < 0.05).

**Table 5 microorganisms-07-00448-t005:** Changes in the color of chicken breasts treated with different chlorine concentrations.

Chlorine (mg/L)	Δ*L*	Δ*a*	Δ*b*	Δ*E*
0	5.4 ± 1.0*^a^*	1.4 ± 0.4*^a^*	3.3 ± 0.9*^a^*	6.5
20	7.0 ± 1.9*^a^*	2.2 ± 0.9*^a^*	1.6 ± 0.7*^a^*	7.5
50	5.6 ± 1.6*^a^*	0.8 ± 0.9*^a^*	2.8 ± 1.6*^a^*	6.3
100	5.5 ± 2.2*^a^*	1.7 ± 1.0*^a^*	2.8 ± 1.4*^a^*	6.4

Mean values followed by different superscript letters are significantly different (*p* < 0.05).

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
