# Peer review of "Modeling the Reduction and Cross-Contamination of Salmonella in Poultry Chilling Process in China"

_microorganisms, 2019, doi:10.3390/microorganisms7100448_

Round 1

Reviewer 1 Report

Xingning Xiao and coworkers in the manuscript entitled: “Modeling the reduction and cross-contamination of Salmonella in poultry chilling process in China presented model for reduction and cross-contamination of Salmonella on chicken in chilling process. For prediction of post-chill incidence, the developed response surface model has a satisfactory performance, which could be used to provide the input for QMRA model of Salmonella in poultry supply chain in China. The manuscript was prepared carefully, using many laboratory techniques that can easily be translated into industry.  For this reason, I recommend this article for publication in present form.

Author Response

General comments:

Xingning Xiao and coworkers in the manuscript entitled: “Modeling the reduction and cross-contamination of Salmonella in poultry chilling process in China presented model for reduction and cross-contamination of Salmonella on chicken in chilling process. For prediction of post-chill incidence, the developed response surface model has a satisfactory performance, which could be used to provide the input for QMRA model of Salmonella in poultry supply chain in China. The manuscript was prepared carefully, using many laboratory techniques that can easily be translated into industry.  For this reason, I recommend this article for publication in present form.

Reply: Thanks for the comments.

Reviewer 2 Report

The manuscript: Modeling the reduction and cross-contamination of Salmonella in poultry chilling process in China by X. Xiao et al. is a very important work, which is a continuation of the research of this group. The work is written at a high level of understanding of statistics and modeling in microbiological research.

In my opinion, after a few corrections it can be published.

I have remark about the numbering of mathematical formulas. Why is Numbers (1) and (1) (2) by the Formula 1 (p 5/14 above line 167) and (4) (3) with formula (4)?

Why is "Ypc" before (%) in the formula (3), and "Ypci" in the formula description (p 5/14 lines 176 and 194)?

I have a problem with point 3.3 in the Results and Discussion chapter. The use of examples should be not based on the result of the test, and on the variable Run, which uniquely assigns the described example, differently than the result that is repeated in the table (40% pre-chill incidence).

Understanding this fragment of the work would be easier if Table 2 was placed not in the chapter Method, but in the Results. I also do not understand why the chlorine concentrations were tested with different number of runs (0mg / L - 1 time; 20-4; 50-10; 80-4, and 100-1) and the same situation was in variable Initial contamination Level (level 1- 1 time; 2-4; 3-10; 4- 4; and 5-1). The table shows that Initial contamination lavel 2 log CFU / g was tested with 20 and 80 mg / L chlorine. Why not with all chlorine concentrations? The most prevalent Initial contamination lavel 3 log CFU / g was tested 1 time with 0mg / L chlorine, 1 time with 100mg / L chlorine, and 8 time with 50mg / L chlorine. I would expect a short explanation for this choice. If such a design was the result of a model developed with the JMP10 software, then this should be described more prominent in the cooling procedure and briefly justify the advantage of such a model.

In addition, superscript letters "abcd" were next to the values of the Post chill incidens column. I completely do not understand the sentence under the table (p4 / 14 line 146), which probably was to clarify their meaning. Does this mean that all values ​​marked with "a" differ significantly? Is the Post-chill incidence variable the average or median of several identical measurements? In this case, Authors would have to specify the number of measurements in a single "Run".

Table 3 is also a research result and a description of the method in one. It lacks a variable that would show the difference between observed and predicted post-chill incidence. This table also requires a brief description of why Initial contamination level 3 and 5 log CFU / g and chlorine 10 and 70mg / L were selected for the study.

Figures 2 and 3 present probability distribution of 2 parameters in chilling water with 20-100mg / L chlorine (Fig.2) and 50-100 mg / L chlorine. Ist it the mean probability distribution of all chlorine concentrations?

In chapter 3.4 Model Development (p8 / 14, line 283-284 the sentence "There were no significant ...." could be worded coherent. Isn't there a correlation between bacterial reduction and: 1) the treatment of 20-100mg / l (chlorine?); 2) treatment time? Or maybe it's describes the comparing of means or medians of the bacterial reduction in different measurement conditions? The phrase "the treatment time was not a significant variable" makes no sense. Treatment time may not affect the measurement of bacterial reduction, but significance assesses differences, correlations, not variables.

Besides, I have no comments.

Best regards

Author Response

General comments:

The manuscript: Modeling the reduction and cross-contamination of Salmonella in poultry chilling process in China by X. Xiao et al. is a very important work, which is a continuation of the research of this group. The work is written at a high level of understanding of statistics and modeling in microbiological research. In my opinion, after a few corrections it can be published.

I have remark about the numbering of mathematical formulas. Why is Numbers (1) and (1) (2) by the Formula 1 (p 5/14 above line 167) and (4) (3) with formula (4)?

Reply: The sentences in line 165 and line 187 have been revised to make the formula number more clearly.

Why is "Ypc" before (%) in the formula (3), and "Ypci" in the formula description (p 5/14 lines 176 and 194)?

Reply: “Ypci” has been revised to “Ypc” in lines 172, 189, 303.

I have a problem with point 3.3 in the Results and Discussion chapter. The use of examples should be not based on the result of the test, and on the variable Run, which uniquely assigns the described example, differently than the result that is repeated in the table (40% pre-chill incidence).

Reply: The sentence has been revised in lines 260-262.

Understanding this fragment of the work would be easier if Table 2 was placed not in the chapter Method, but in the Results. I also do not understand why the chlorine concentrations were tested with different number of runs (0mg / L - 1 time; 20-4; 50-10; 80-4, and 100-1) and the same situation was in variable Initial contamination Level (level 1- 1 time; 2-4; 3-10; 4- 4; and 5-1). The table shows that Initial contamination lavel 2 log CFU / g was tested with 20 and 80 mg / L chlorine. Why not with all chlorine concentrations? The most prevalent Initial contamination lavel 3 log CFU / g was tested 1 time with 0mg / L chlorine, 1 time with 100mg / L chlorine, and 8 time with 50mg / L chlorine. I would expect a short explanation for this choice. If such a design was the result of a model developed with the JMP10 software, then this should be described more prominent in the cooling procedure and briefly justify the advantage of such a model.

Reply: As suggested, Table 2 was moved to the chapter Results. In this study, response surface methodology was applied with central composite design. This design led a single block of 20 sets of test conditions containing points of factorial design, axial points and central points. All factors are studied in five levels (-a, -1, 0, +1, +a) and a-values depend on the number of variables. For two, three, and four variables, they are, respectively, 1.41, 1.68, and 2.00. The graphical representation for three factors of central composite design can be illustrated in Figure 1. Each point represents a set of test conditions in central composite design.

Figure 1 Central composite designs for the optimization of three variables (a = 1.68). (●) Points of factorial design, (○) axial points and (□) central point.

The description has been added in the lines 124 to 134 to explain the advantage of the central composite design as “In multivariable analyses, the traditional optimization technique of changing one variable at a time to study the variable response effect is impracticable and does not represent the interaction effect between different factors. Therefore, experimental statistical design considers one of the most useful methods in obtaining valuable and statistically significant models of a phenomenon by performing a minimum number of calculated experiments. It also considers interactions among the variables and can be used to optimize the operating parameters in multivariable analyses. Response surface methodology was used for the modeling and analysis of problems in which a response of interest is influenced by several variables to optimize the same response. A favorite model in response surface methodology is the central composite design, which is efficient and flexible in providing adequate data on the effects of variables and overall experiment error even with a fewer number of experiments”.

Saeed, M.O.; Azizli, K.; Isa, M.H.; Bashir, M.J.K. Application of ccd in rsm to obtain optimize treatment of pome using fenton oxidation process. Journal of Water Process Engineering,2014, 7-16.

Mohana, S.; Shrivastava, S.; Divecha, J.; Madamwar, D. Response surface methodology for optimization of medium for decolorization of textile dye direct black 22 by a novel bacterial consortium.Bioresource Technol, 2008, 99, 562-569.

Bezerra, M.A.; Santelli, R.E.; Oliveira, E.P.; Villar, L.S.; Escaleira, L.A. Response surface methodology (RSM) as a tool for optimization in analytical chemistry. Talanta, 2008, 76, 965-977.

In addition, superscript letters "abcd" were next to the values of the Post chill incidens column. I completely do not understand the sentence under the table (p4 / 14 line 146), which probably was to clarify their meaning. Does this mean that all values ​​marked with "a" differ significantly? Is the Post-chill incidence variable the average or median of several identical measurements? In this case, Authors would have to specify the number of measurements in a single "Run".

Reply: As described in lines 141 to 143, the whole experiment was repeated twice on different days and duplicated plates were used in microbial tests for each chicken breast sample, so the post-chill incidence values in Table 2 were the means ± standard deviations (n = 2). It has been specified in line 273. Values followed by different superscript letters are significantly different (p < 0.05), for example, when one value has the superscript letters "abcd" and the other value has the superscript letters “a” or “b” or “c” or “d” letters, it means there were no significant between two values.

Table 3 is also a research result and a description of the method in one. It lacks a variable that would show the difference between observed and predicted post-chill incidence. This table also requires a brief description of why Initial contamination level 3 and 5 log CFU/ g and chlorine 10 and 70mg / L were selected for the study.

Reply: The trails showed in Table 3 were used for model validation with the bias factors (Bf) and accuracy factors (Af), which showed the difference between observed and predicted post-chill incidence (lines 195-196). The principle of the model validation variable is the selected parameters were within the original range of the experimental design but not included in the development of the model (lines 196-197). In this study, the verification experimental group with initial contamination level (3 and 5 log CFU/g), pre-chill incidence (12.5% and 37.5%) and chlorine concentrations (10 and 70 mg/L) were within the original range of the experimental design but not included in the development of the model.

Wang, W.; Li, M.; Fang, W.; Pradhan, A. K.; Li, Y. A predictive model for assessment of decontamination effects of lactic acid and chitosan used in combination on Vibrio parahaemolyticus in shrimps. Int J Food Microbiol, 2013, 167, 124-130.

Figures 2 and 3 present probability distribution of 2 parameters in chilling water with 20-100mg / L chlorine (Fig.2) and 50-100 mg / L chlorine. Ist it the mean probability distribution of all chlorine concentrations?

Reply: The bacterial reductions in 20-100 mg/L chlorine within 40 min were not significant correlated to the chlorine concentration and the treatment time, so the probability distribution in Figure 2 described the range and the uncertainty feature of the bacterial reductions with 20-100 mg/L of chlorine. As the same, the probability distribution in Figure 3 described the range and the uncertainty feature of bacterial transfer rates with 50-100 mg/L of chlorine.

In chapter 3.4 Model Development (p8 / 14, line 283-284 the sentence "There were no significant ...." could be worded coherent. Isn't there a correlation between bacterial reduction and: 1) the treatment of 20-100mg / l (chlorine?); 2) treatment time? Or maybe it's describes the comparing of means or medians of the bacterial reduction in different measurement conditions? The phrase "the treatment time was not a significant variable" makes no sense. Treatment time may not affect the measurement of bacterial reduction, but significance assesses differences, correlations, not variables.

Reply: As suggested, the sentence has been revised to “There were not significant correlation between bacterial reductions with the 20-100 mg/L chlorine treatments and the chilling time (p > 0.05)” in lines 276-277 and in lines 281-283, “Besides, there were not significant correlation between bacterial transfer rates with the treatments of 50-100 mg/L of chlorine and the chilling time (p > 0.05)”

Other revisions

The citation and references have been revised accordingly.